Disentangling the link between supplemental feeding, population density, and the prevalence of pathogens in urban stray cats

http://orcid.org/0000-0002-9628-9595 Hwang Jusun 1 2
Gottdenker Nicole L. 2
Oh Dae-Hyun 1
Nam Ho-Woo 3
Lee Hang 1
Chun Myung-Sun 1 jdchun@snu.ac.kr
1 The Research Institute for Veterinary Science, College of Veterinary Medicine, Seoul National University , Seoul , Republic of Korea
2 Department of Veterinary Pathology, University of Georgia , Athens, GA , USA
3 Parasitic Disease Research Institute, College of Medicine, Catholic University of Korea , Seoul , South Korea
Bienzle Dorothee
Electronic publication date: 2018 Jun 25
Publication date: 2018
Volume: 6
Electronic Location ID: e4988
Received 2018 Mar 7; Accepted 2018 May 26
Copyright: © 2018 Hwang et al.
Copyright year: 2018
Copyright holder: Hwang et al.
License: This is an open access article distributed under the terms of the Creative Commons Attribution License, which permits unrestricted use, distribution, reproduction and adaptation in any medium and for any purpose provided that it is properly attributed. For attribution, the original author(s), title, publication source (PeerJ) and either DOI or URL of the article must be cited.
License URL: https://creativecommons.org/licenses/by/4.0/

Keywords: Feline pathogen, Supplemental feeding, Host population density, Urban stray cat, Urban ecology, Cat caretaker activity

Funding: National Research Foundation of Korea NRF-2012S1A5B6034265 This work was supported by the National Research Foundation of Korea (Grant Number: NRF-2012S1A5B6034265). The funders had no role in study design, data collection and analysis, decision to publish, or preparation of the manuscript.

==============================
Background

Supplemental feeding of free-roaming animals, including wildlife and feral or stray animals, is well known to have a substantial impact on various aspects of animal ecology including habitat use, activity patterns, and host-pathogen interactions. Among them, an increased population density (PD) of animals receiving supplemental food raises concerns regarding the transmission of pathogens in these host populations. The primary aim of this study was to investigate how supplemental feeding is associated with host PD and prevalence of pathogens with different transmission modes in urban stray cats. We hypothesized that supplemental feeding would be positively associated with host PD and the prevalence of pathogens with density-dependent transmission modes compared with pathogens with transmission modes that are considered relatively density-independent.

Methods

This study was conducted in six districts in Seoul, Republic of Korea which were selected based on different degrees of supplemental feeding and cat caretaker activity (CCA). The PD of stray cats was estimated by mark-recapture surveys. Stray cat blood samples (N = 302) were collected from stray cats by local animal hospitals from each district performing the trap-neuter-release which tested for eight pathogens with different transmission modes (feline immunodeficiency virus, feline leukemia virus (FeLV), feline panleukopenia virus, feline calicivirus, feline herpesvirus-1, Bartonella henselae, hemoplasma, and Toxoplasma gondii) with molecular or serological assays. Associations between the prevalence of each pathogen and PD, CCA, and sex of cats were statistically analyzed.

Results

In contrast to initial predictions, the cat PD was generally higher in low CCA districts. The prevalence of (FeLV), which is transmitted through direct contact, was significantly higher in areas with a high CCA, conforming to our hypothesis. On the other hand, the prevalence of feline parvovirus, which can be spread by environmental transmission, was higher in low CCA districts. The remaining six pathogens did not show any association with the CCA; however, they had a unique association with the PD or the sex of the stray cats.

Discussion

Our findings suggest that in addition to influencing the PD, supplemental feeding may affect the prevalence of pathogens in urban animals by mechanisms such as increased aggregation and/or altered foraging strategies, with different consequences depending on the transmission mode of each pathogen.

Introduction

Supplemental feeding of free-ranging animals is a common practice worldwide that occurs through multiple direct (e.g., feeding for conservation management and backyard bird feeders) or indirect (e.g., urban trash cans and pet’s food outdoors) routes (Oro et al., 2013; Becker, Streicker & Altizer, 2015; Murray et al., 2016). However, an increasing number of studies have raised concerns regarding the impact of anthropogenic food supplementation on various aspects of animal ecology, ranging from individual phenotypic traits (e.g., body size and immunocompetence), population dynamics (e.g., survival rate, breeding rate and population density (PD)), to pathogen host community composition (Robb et al., 2008; Rodriguez-Hidalgo et al., 2010; Jessop et al., 2012; Galbraith et al., 2015). For instance, increased body mass, PD, and reproduction rates were observed in red deer (Cervus elaphus) that were provided with artificial food sources in effort to maintain their population size as a game species (Vicente et al., 2007; Rodriguez-Hidalgo et al., 2010). These accumulated changes in individual biology and population dynamics are suspected to have further and unexpected consequences on disease transmission because the dependence of animals on supplemented food can ultimately alter their exposure and/or susceptibility to pathogens. Among diverse mechanisms linking supplemental feeding and altered host-pathogen interactions, a heightened PD (increase in general abundance of animals in a certain area) and crowding (spatiotemporal aggregation of animals) of host individuals has been frequently discussed for several reasons. For instance (1) increased stress of the animals as a result of crowding may lead to negative influences on the immune function of animals and their susceptibility of pathogen infection (Semeniuk & Rothley, 2008; Forristal et al., 2012), while (2) increased frequency of close contact between infectious and susceptible host individuals around the food source may mechanically accelerate the transmission of pathogens. This has been particularly well documented in avian species regarding backyard bird feeders, where the crowding of birds contributes to the spread of mycoplasma conjunctivitis, caused by Mycoplasma gallisepticum (Adelman et al., 2015; Moyers, 2017).

The influence of supplemental feeding on host-pathogen interactions as a result of increased host PD can further be complicated because the response of a pathogen to host density may vary based on its unique transmission mode. Theoretically, pathogen transmission dynamics are commonly categorized based on different assumptions about the dependence of pathogen transmission on PD of host populations. Pathogens transmission through contact that increases proportionately with PD, is regarded as “density-dependent transmission,” whereas when pathogen transmission is driven by the “frequency of infected contacts,” independent of host density, it is identified as “frequency-dependent transmission.” For pathogen groups adopting density-dependent transmission, its transmission is facilitated as host PD increases and the chances of individuals having contacts in close proximity becomes frequent, as with Brucella abortus in ungulates. (Dobson & Meagher, 1996). However, pathogens with frequency-dependent transmission modes, such as vector-borne and sexually or trophically-transmitted pathogens, often depend on specific behaviors of the infectious vectors or host individuals, such as aggressive behavior during the breeding season or exposure to hematophagous arthropods, which enables the transmission of the pathogens, rather than PD or physical contact in proximity among host individuals (Thrall, Antonovics & Hall, 1993). In relation to supplemental feeding, increased host aggregation in response to a spatiotemporally stable food source may increase the transmission of directly transmitted pathogens as observed in the increased prevalence of endoparasites in raccoons (Procyon lotor) with altered contact rates (Wright & Gompper, 2005). However, it is possible that such a pattern may not apply to other pathogens as observed in the cases of decreased prevalence of echinococcosis in urban red foxes (Vulpes vulpes) compared to those in rural area, potentially due to their reduced predation on intermediate hosts in urban habitats (Fischer et al., 2005). However, to date, few studies have simultaneously evaluated the linkages between supplemental feeding and the prevalence of pathogens with differing transmission modes (Miller et al., 2003; Vicente et al., 2007; Brennan et al., 2014).

Modifications of host-pathogen interactions due to supplemental feeding may also apply to other free-roaming animals. Among them, domestic cats (Felis catus) are one of the species that have managed to prosper in a wide range of habitats, including well developed cities (Bateman & Fleming, 2012). Urban free-roaming cats can benefit from surplus food sources provided by urban residents. These residents who regularly or sporadically provide a supplemental food source for free-roaming cats are commonly referred to as cat caretakers (Centonze & Levy, 2002; Finkler, Hatna & Terkel, 2011b; Peterson et al., 2012). There are many ecological and sociological issues centered around free-roaming cats, as the availability of predictable food sources and a stable energy intake are known to increase the overall abundance of supplemented animals (López-Bao et al., 2010; Rickett et al., 2013; Plummer et al., 2013). Among them, the epidemiological role of urban free-roaming cats regarding public health and the health of other domestic and wild animals is a subject of concern (Lepczyk, Lohr & Duffy, 2015). The most well-known zoonotic pathogens and diseases infecting cats include rabies, Toxoplasma gondii, and various arthropod-borne pathogens such as Bartonella henselae and Rickettsia species (Gerhold & Jessup, 2013; Spada et al., 2014).

The aim of this study was to examine how human food provisioning, host PD, and pathogen prevalence are associated in urban stray cats while asking how these relationships vary with the transmission modes of each pathogen. Here, we defined the group of unowned and free-roaming cats in a city as urban stray cats. We investigated the prevalence of eight feline pathogens, feline calicivirus (FCV), feline herpesvirus-1 (FHV-1), feline leukemia virus (FeLV), feline immunodeficiency virus (FIV), feline hemoplasma, B. henselae, feline parvovirus (FPV), and T. gondii. Each of these pathogens has a different transmission mode, falling somewhere along the continuum of density-dependent and frequency-dependent transmission (Table 1) (Fenton et al., 2002; Begon et al., 2002; Greer, Briggs & Collins, 2008). FCV, FHV-1, and FeLV are pathogens with density-dependent transmission modes because its transmission is commonly facilitated by close contact in proximity driven by aggregation and crowding of individuals, such as through aerosols (Radford et al., 2009; Thiry et al., 2009). In contrast to FIV, feline hemoplasma, and B. henselae are either transmitted during reproductive behavior (FIV and feline hemoplasma) (Courchamp, Say & Pontier, 2000; Willi et al., 2007), or through arthropod vectors (B. henselae and feline hemoplasma) (Chomel et al., 2006; Willi et al., 2007). Hence, the transmission modes of these three pathogens are assumed to be relatively frequency-dependent (Thrall, Antonovics & Hall, 1993). Lastly, FPV and T. gondii are pathogens mostly transmitted through environmental contaminants rather than direct contact between infectious and susceptible individuals. As environmental feces concentration, which function as common point source of exposure for environmentally-transmitted pathogens, is expected to increase in relation to PD, we also assume the positive association between the transmission of FPV and T. gondii with host PD. In summary, we hypothesize that supplemental feeding would increase the PD of stray cats, leading to heighted prevalence of pathogens transmitted through direct contact with infectious individuals or with environmental contaminants (density-dependent transmission: FCV, FHV-1, and FeLV; environmental transmission: FPV and T. gondii) (Lutz et al., 2009; Radford et al., 2009; Thiry et al., 2009; Ballash et al., 2015). Meanwhile we expect that the prevalence of sexually-transmitted, vector-borne, or trophically-transmitted pathogens (frequency-dependent transmission: T. gondii, FIV, hemoplasma, and B. henselae) would be relatively unaffected by difference in host PD and supplemental feeding (Courchamp, Say & Pontier, 2000; Willi et al., 2007; Chomel et al., 2009; Afonso et al., 2013).

Table 1 Transmission modes of studied pathogens.

Pathogens	Transmission modes	Relative position of pathogens on continuum of transmission modes	
Feline calicivirus (FCV)	Aerosol in close proximity, direct contact with contaminated objects or infected cats	Density-dependent transmission
Frequency-dependent transmission	
Feline herpesvirus (FHV)	Aerosol in close proximity, direct contact with contaminated objects or infected cats	
Feline leukemia virus (FeLV)	Saliva, aerosol in close proximity	
Toxoplasma gonii	Prey consumption contact with contaminated abiotic materials (e.g., feces)	
Feline parvovirus (FPV)	Saliva, blood, urine, and feces	
Bartonella henselae	Vector-borne, mostly flea	
Hemoplasma	Through saliva and/or arthropod vectors	
Feline immunodeficiency virus (FIV)	Aggressive behavior during breeding seasons-saliva and bite wounds	
Note:

Specific transmission routes of studied pathogens and its relative position along the continuum between density-dependent and frequency-dependent transmission.

Materials and Methods

Study area

The study was performed in Seoul, the capital city of the Republic of Korea (ROK). Seoul is one of the largest megacities in the world with a population size and density of approximately 9,908,000 and 17,000 people per km2, respectively (Kim & Baik, 2005). The city has a temperate climate with four seasons and an average temperature of 13.2 °C in the spring and fall, 24.3 °C in the summer and −2.4 °C in the winter. Cat ownership is rapidly increasing in the ROK, and according to a recent national survey, cat ownership increased from 1.7% in 2010 to 3.4% in 2012 and to 5.2% in 2015 (Animal and Plant Quarantine Agency, 2015). Most owned cats in Seoul are kept indoors due to hygienic reasons and the fear of mortality from vehicle collisions. Hence, free-roaming cats in the city are commonly considered as unowned, stray cats. In the city, stray cats are most commonly observed in residential areas including apartment complexes and intensive housing areas.

Seoul is divided into 25 administrative local districts. We ranked the districts based on the intensity of the cat caretaker activity (CCA) of each district. To rank all 25 districts based on the intensity of the CCA, we selected parameters that reflect the following features: spatiotemporal regularity of the feeding behavior, socioeconomic characteristics related to the cat feeding behavior, the number of cat caretakers and the number of cats cared for per caretaker. In total, we used eight parameters, and information on six of those parameters were extracted from the result of a 2013 nationwide cat-caretaker questionnaire (Seoul; n = 1,048) (Kim et al., 2016) as follows: (1) proportion of survey respondents who identified themselves as cat caretakers in the total population of each district, (2) proportion of respondents who are taking care of more than 10 cats, (3) proportion of respondents who have been working as a cat caretaker for more than five years, (4) proportion of respondents who provide food supplement daily in a regular manner, (5) proportion of respondents who provide food in areas further than 100 m radius from their house, and (6) a score from the subjective perception of each respondent about the intensity of the CCA in his/her residing district. In addition, we included two more demographic factors for each district, (7) matriculation rates, and (8) property tax (index of wealth), which are known to be positively associated with the intensity of the CCA in a previous study (Finkler, Hatna & Terkel, 2011b). Second, we performed a principal component analysis of these eight parameters. The first principal component (PC1) explained 39% of the variance in the data (Table 2), and all the parameters explaining the intensity of the CCA correlated positively with the PC1. Hence, the PC1 was used as the index reflecting intensity of the CCA and the 25 city districts were ranked based on this index (Fig. 1).

Table 2 Principal components (PC1 and PC2) eigenvalues and PC loadings for the CCA parameters.

	PC1	PC2	
Eigenvalue	3.12	1.41	
Percentage variation explained	0.39	0.18	
Eigenvectors			
 Matriculation rate	0.45	−0.27	
 Property tax	0.46	−0.11	
 Proportion of survey respondents	0.37	−0.42	
 Proportion of respondents taking care of more than ten cats	0.24	0.58	
 Proportion of cat caretakers with more than five years of experience	0.29	0.39	
 Provide food supplement daily in a regular manner	0.27	0.41	
 Provide food in areas further than a 100-m radius from their house	0.25	−0.27	
 Subjective score about the intensity of the CCA in his/her residing district	0.42	0.08	

Figure 1 CCA (PC1) in the districts of Seoul based on the PCA result.

The six districts selected are; GN, SC, and MP (high CCA districts; yellow), and GC, DDM, and SD (lower CCA districts; blue). District SP and GD was excluded from the study site due to logistical reasons.

Out of eight districts with highest or lowest CCA based on the ranking (Fig. 1), we excluded two districts, one with high CCA (Songpa; SP) and the other with low CCA (Gangdong; GD), for different reasons. SP is a unique district extensively consisting of many apartment complexes where each apartment complex has their own specific policy on how to deal with stray cats (trapping and euthanizing, poisoning, active CCA, etc.). Therefore, we assumed it would be inappropriate to compare cat populations from SP with other districts; it was excluded from this study. On the other hand, district GD was excluded due to unique local condition regarding stray cats. Although GD scored low in our CCA index, the district office and cat caretakers of GD have been working in cooperation to systemically provide food for stray cats in its district for years. For instance, they established various official feeding sites within the district for stray cats. Hence, despite the low CCA of GD, the feeding ecology of stray cats within the district may differ from other districts with similar CCA and were excluded from this study. Hence, six remaining districts were used as our study sites, three with a high CCA (Gangnam; GN, Seocho; SC, Mapo; MP) and three with a low CCA (Geumcheon; GC, Seongdong; SD, Dongdaemoon; DDM) (Fig. 1). Hereafter the three high CCA sites will be labeled as GN (H), SC (H), and MP (H), and three low CCA sites as GC (L), SD (L), and DDM (L).

Field surveys

We performed PD surveys at 12 sites from the six selected districts (two sites were selected in each district) (Fig. 2). Due to logistical reasons, a survey of four sites (two high CCA sites from district SC (H), and two low CCA sites from district GC (L)) was conducted in the fall (September to November) of 2014 and the rest of the sites in the spring (April to June) of 2015. At each site, one survey transects of 2 km was established from the residential areas within each district. Apartment complexes were excluded from the survey routes to prevent sampling bias because different apartment complexes tend to have specific stray cat management policies. Each transect was surveyed five times, once per day for five days, all within two weeks period. Typical residential areas in the city used for the density survey consisted of rows of multistory houses (usually 2–4 stories) constructed parallel to each other with a narrow alley in between (Fig. S1). At each site, the transect routes were selected in a zig-zag configuration to thoroughly cover the blocks, as in methods used in previous studies (Calhoon & Haspel, 1989; Finkler, Hatna & Terkel, 2011a). Alleys were walked by the same person throughout the survey (JH). Surveys were started within one hour from sunset and usually completed within 60–70 min. These times were selected based on the nocturnal characteristics of the cat activity (Izawa, 1983; Horn et al., 2011). All the cats observed in the alleys were recorded as well as the ones on the roofs and walls of houses on the both sides of the alleys. Due to the dense structure of the houses in these area, the lateral view from the alleys was limited, and in most cases, the observer was only able to see the walls and the roof top of the houses right next to the alleys. Only cats spotted near the observer, approximately within 2 m of range, close enough to identify its full morphological characters, were counted and recorded. For every cat observed, the location, time of observation, and unique characteristics (morphological features, behavior toward survey personnel, health status based on external observation, and sex when possible) were recorded. Individual cats were primarily identified by their coat color. A detailed coat pattern was depicted at the observed site on a record sheet printed with outlines of cat body forms in various angles and postures. Cats with completely uniform color throughout their body were very rare (eight out of 277 total observed cats), mostly observed at different sites and were identified by other characteristics such as body size, tail shape, or hair length. Data on the sterilization of the cats were identified and recorded by the ear-cut mark made during the sterilization procedure. In Seoul, sterilized stray cats are marked with a straight cut of the ear-tip, rather than a notch, which is easily identified by its abrupt shape. During this survey, we did not observe any cat with physical scars that would be confused with the ear mark. Kittens that were accompanied by their mother were not included in the density analysis. We did not observe any kittens roaming by themselves during our surveys.

Figure 2 Map of Seoul showing the 6 districts where the study was performed1.

High CCA districts in red letters and low CCA districts in blue letters.

Biological sample collection

Biological samples were collected from the six study districts (GN, SC, MP (high CCA) and GC, DDM, and SD (low CCA)) from September, 2014 to June, 2015 except for the cold season (from December, 2014 to March, 2015) when trap-neuter-release (TNR) is not done. The time frame of the sample collection overlapped with that of the field density survey. The Seoul Metropolitan Government has been performing TNR since 2008 as part of official city service. In each district, designated animal hospitals perform TNR of stray cats under a contract with the Seoul city government. We cooperated with animal hospitals from the six districts we selected and requested that they collect 50 adult cat blood samples per district during the sterilization procedures (Yeonnam Animal Hospital, MP (H); Hanbit Animal Hospital, SD (L); Kang Hyunrim Animal Hospital, GC (L); Gangnam 24hr Animal Hospital, GN (H); Han Animal Hospital, SC (H), and Jeonnong Animal Hospital, DDM (L)). All blood samples were drawn after the sterilization surgery by certified veterinarians at each animal hospital. The protocol explaining the preferred site and volume for the blood draw (jugular vein) and the data sheet to input the types of information required for this study were provided to the veterinarians in advance with the syringes and tubes required for our sample collection. All procedures were performed under full anesthesia of the animal with a combination of Ketamine (22 mg/kg) and Xylazine (1.1 mg/kg) injected through the hind leg muscle. All the cats that were determined to be yearlings or younger by the veterinarians and were released without being sterilized; hence, no samples were collected from these individuals.

Half of each blood sample (approx. 1.5–2 ml) was placed into a tube containing the anticoagulant ethylene diamine tetra-acetic acid (Becton Dickinson, Rutherford, NJ, USA), and the other half was placed into a serum separator tube. Blood samples collected for serum separation were permitted to clot and then centrifuged for 10 min at 1,200 relative centrifugal force. Next, the serum was removed from the clot using a sterile pipet and transferred to a sterile 1.5 ml Eppendorf tube. All serum and whole blood samples were stored at −70 °C until analyses were conducted.

Laboratory analysis

Polymerase chain reaction (PCR) was used to identify feline hemoplasma in the whole blood samples. Frozen whole blood samples were thawed at room temperature and genomic DNA was extracted using the QIAamp DNA mini kit (Qiagen, Hilden, Germany). The extracted DNA was used as a template for the subsequent PCR assay. We amplified a partial segment of the 16S rRNA gene of mycoplasma species (including hemoplasma, which is the hemotropic mycoplasma) using the universal primer set HBT-F (ATACGGCCCATATTCCTACG) and HBT-R (TGCTCCACCACTTGTTCA) described in a previous study (Criado-Fornelio et al., 2003). Every PCR run included a positive and negative control to detect bacterial contamination. Amplified products were separated by electrophoresis on a 1.2% agarose gel. PCR amplicons from the positive samples were purified and prepared for direct sequencing. The sequence homology of the sequenced 16S rRNA gene from the positive samples was examined using the BLAST software (National Center for Biotechnology Information; available at: http://www.ncbi.nlm.nih.gov).

Serological assays were performed to determine the prevalence of T. gondii, B. henselae, FIV, FeLV and FCV, FHV-1, and FPV in the samples. Because different serological tests were not performed on the same day, aliquots of serum samples were stored in several tubes so that each tube was thawed for a single analysis only to prevent repeated thawing and freezing of the samples. Commercial ELISA test kits were used to detect (1) antibodies for FIV and FeLV p27 antigen detection (SensPERT FeLV Ag/FIV Ab kit; VetAll™) and (2) antibodies for T. gondii rapid diagnostic test developed by Chong et al. 2011, using recombinant SAG1 antigen.). Tests to detect B. henselae antibodies were done with the indirect immunofluorescence (IFA) slides test (B. henselae IFA feline IgG antibody kit; Fuller Laboratories, Fullerton, CA, USA). A titer of >1:64 IgG immunoglobulins was considered positive for exposure to B. henselae. Lastly, modified in-house ELISA kits (ImmunoComb, Feline VacciCheck, Biogal Galed Laboratories, Gal’ed, Israel) were used to test for FCV, FHV-1, and FPV antibodies. The ImmunoComb® test is based on the solid phase “dot”-ELISA technology consisting of a comb-shaped plastic card. On each of the 12 teeth on the plastic card, there are four spots as follows: FCV antigen is the lowest spot; FHV-1 antigen is next to the lowest spot; FPV antigen is the second spot from the top; and the uppermost spot is the positive reference. The titer of the positive index used to test for antibodies to each pathogen in ImmunoComb was as follows: FPV antibodies at 1:80 in the hemagglutination-inhibition (H.I.) reaction; FHV-1 antibodies at 1:16, and FCV antibodies equal to 1:32 in the virus neutralization reactions. Tests for all antigens and antibodies were performed based on the protocols provided by the manufacturers.

Estimating population density

The PD estimation was performed by counting of individual cats identified by their morphological characteristics (Harihar et al., 2009; Finkler, Hatna & Terkel, 2011a). Information of morphological characteristics were primarily recorded during direct observation, but also collected by photographs taken during the survey. We used a capture–recapture sampling technique: there was an initial capture during which each observed cat was individually identified and recorded, and four consecutives “recapture” surveys in which re-sighted cats were counted as “recapture.” Population estimation was done based on the closed population model assuming there was limited immigration/emigration, birth, and death during the two-week survey period (Gross et al., 2012). Using the program MARK (Cooch & White, 2001), we built four models, M0, Mt, Mh, and Mth, with different variables and assumptions. M0 is the null model with the principal assumption that the probability of capturing an animal is constant over all animals over all periods, while the heterogeneity model, Mh, permits the capture probability to vary for each individual. Mt assumes each animal has a constant capture probability on any sampling occasion; however, the probability of capture can vary from one occasion to the next (Seber, 1992). Finally, model Mth simultaneously assumes the heterogeneity in the capture probability among individuals and between sampling occasions. Another trap response model commonly used is Mb which captures the variation of the individual response to previous trapping (e.g., trap-shy and trap-neutral). However, this model was not used in our analysis because our survey was based on capturing cats visually rather than physical trapping, and we did not consider our visual observation as a likely source of behavioral variation. We evaluated the relative fit of the models using the sample-size-adjusted Akaike information criterion (AICc) index (Burnham & Anderson, 1998). The model with the best fit was selected based on its AICc values, delta AICc, and AICc weight. For each site, the population size was obtained as the derived parameter of the best fit model based on the AICc model selection. The total number of individuals observed at each site was multiplied by the size of the survey area to obtain estimates of the cat density (per km2) and 95% confidence intervals. Survey areas were determined in ArcView 9.0 (ESRI, Redlands, CA, USA) by creating a polygon with a 50-m buffer zone around each survey transect. The length of the buffer zone was based on the result from a recent study on the home range of urban free-roaming cats using GPS tracker, where the smallest home range size was 0.99 ha. (Thomas, Baker & Fellowes, 2014). This home range being equivalent to a circular area of dimeter of approximately 55 m, we used 50 m as our buffer zone as a conservative estimate. The resulting size of the survey sites ranged from 0.13 to 0.20 km2. The PD estimated from two sites within each district was averaged to enable the analysis of each district’s association with the prevalence of tested pathogens.

Statistical analysis

All statistical analyses were performed with R (R Core Team, 2017). Chi-square tests were done to ensure that there was no sex bias in the blood sample size among the six districts. The normality of the PD of the 12 survey sites were confirmed through Shapiro–Wilk tests. Two linear mixed-models (LMMs) were used to test the associations between the CCA and the PD or the sterilization rates. In each model, the PD or the sterilization rates were set as a response variable. In both models, the CCA was set as the explanatory variable, and “district” and “year (density survey done in 2014 or 2015)” were set as random effects. We also tested for the presence of associations between the prevalence of each pathogen (response variable) and sex, PD, and CCA using generalized linear models with the logit link function. None of the interaction factors showed a significant effect on the response variable (prevalence of each pathogen) and were not added to the analysis. For the above statistical analyses, we considered a p-value of ≤ 0.05 as significant. We used a model selection approach (AICc) to determine the best model or suite of models (Burnham & Anderson, 1998). For each pathogen, we did model-averaging for most parsimonious models (delta AICc smaller than two units) to further analyze the estimates of the model parameters. Analyses were performed using R package “AICmodav” and “lme4” (Bates et al., 2015; Mazerolle, 2017). Lastly, we calculated the mean and standard deviation of the stray cat sterilization rates in studied districts.

Results

Across the 12 survey sites we observed a total of 276 cats during 60 surveys. The estimated number of cats at each survey site ranged from 132 individuals to 268 individuals per km2, with mean 186.8 per km2 (Table 3). However, two replicate sites from the same district showed a similar number of estimated cats, although in two districts (DDM (L), GC (L)), the gap between the replicate sites within each district was rather large (DDM (L) 1 and DDM (L) 2 had 132 and 175 individuals/km2; GC (L) 1 and GC (L) 2 had 168 and 231 individuals/km2), implying a possible heterogeneity in the abundance of stray cats within the districts (Fig. 3). There was no association between the PD and CCA (LMM; Wald χ2 = 1.06, p = 0.30) although trends for a higher PD in low CCA districts (GC (L), SD (L)) and a lower PD in high CCA districts (GN (H), MP (H)) were observed. Based on the observation of the TNR ear marking, the proportion of sterilized cats ranged from 0 to 0.68 with overall average of 0.23. High CCA districts had a significantly higher proportion of sterilized cats (0.36 ± 0.22), compared to that of the low CCA districts (0.10 ± 0.09) (LMM; Wald χ2 = 7.34, p < 0.01; Table 3).

Table 3 Estimated population densities (95% confidence intervals) and proportion of observed cats with the sterilization mark from the six districts (CCA indicated in parenthesis; high-H, low-L).

District (CCA)	Site ID	Area (km2)	Estimated cat density (95% CI cats per km2)	Proportion of sterilized cats*	
GN (H)	GN1	0.14	155 (127–246)	0.24	
GN2	0.17	168 (132–288)	0.68	
SC (H)	SC1	0.19	185 (151–295)	0.30	
SC2	0.17	172 (129–312)	0.22	
MP (H)	MP1	0.15	166 (146–232)	0.14	
MP2	0.18	182 (155–280)	0.58	
DDM (L)	DDM1	0.18	132 (126–172)	0.19	
DDM2	0.13	175 (152–266)	0.00	
GC (L)	GC1	0.19	189 (158–289)	0.22	
GC2	0.16	206 (167–328)	0.08	
SD (L)	SD1	0.18	243 (209–349)	0.00	
SD2	0.21	268 (189–466)	0.10	
Note:

* Calculated as “number of observed cats with an ear mark/number of total observed cats.”

Figure 3 Cat population density estimates (95% confidence intervals) from the 12 survey sites.

Blood samples collected for pathogen prevalence testing showed no sex bias among the six districts (χ2 = 7.25; p = 0.20; d.f. = 5). The overall prevalence of the pathogens was relatively high for FCV (94.3% (282/299)), FHV-1 (97.9% (293/299)), and FPV (79.2% (236/298)) but low for the rest of the pathogens: T. gondii, 7.7% (23/298); FIV, 3.6% (11/302); FeLV, 23.2% (70/302), B. henselae, 34.8% (105/302); and feline hemoplasma, 32.5% (98/302). Among them, the FeLV prevalence was highest in GN (H) and SC (H), both high CCA districts, whereas the FPV prevalence was highest in DDM (L) and SD (L), followed by GC (L) which were the three low CCA districts (Table 4).

Table 4 Prevalence (%), 95% confidence interval and sample size (n)) of each pathogen from the stray cats in the six studied districts (Letter inside the parentheses indicates the CCA of each district; H-high, L-low).

District	Prevalence (%), 95% confidence interval and sample size (n)) of each pathogen	
FIV	Feline Hemoplasma	B. henselae	T. gondii	FPV	FeLV	FCV	FHV-1	
GN (H)	0% (0.00–0.05, n = 50)	26.0% (0.15–0.39, n = 50)	42.0% (0.29–0.56, n = 50)	6.0% (0.02–0.15, n = 50)	59.0% (0.45–0.72, n = 49)	42.0% (0.28–0.56, n = 50)	100% (0.95–1.00, n = 50)	100% (0.95–1.00, n = 50)	
SC (H)	6.0% (0.02–0.15, n = 50)	38.0% (0.26–0.52, n = 50)	30.0% (0.19–0.44, n = 50)	6.0% (0.02–0.15, n = 50)	76.0% (0.63–0.86, n = 50)	36.0% (0.24–0.50, n = 50)	92.0% (0.82–0.97, n = 50)	94.0% (0.85–0.98, n = 50)	
MP (H)	6.0%, (0.02–0.15, n = 50)	48.0% (0.35–0.62, n = 50)	42.0% (0.29–0.56, n = 50)	6.3% (0.02–0.16, n = 48)	75.5% (0.64–0.87, n = 49)	12.0% (0.03–0.21, n = 50)	98.0% (0.91–1.00, n = 49)	96.0% (0.88–0.99, n = 49)	
DDM (L)	4.0% (0.01–0.12, n = 50)	28.0% (0.17–0.41, n = 50)	32.0% (0.20–0.46, n = 50)	4.0% (0.01–0.12, n = 50)	94.0% (0.85–0.98, n = 50)	20.0% (0.11–0.33, n = 50)	100% (0.95–1.00, n = 50)	100% (0.95–1.00, n = 50)	
GC (L)	4.0% (0.01–0.12, n = 50)	28.0% (0.17–0.41, n = 50)	34.0% (0.22–0.48, n = 50)	12.0% (0.05–0.23, n = 50)	80.0% (0.67–0.89, n = 50)	14.0% (0.07–0.26, n = 50)	92.0% (0.82–0.97, n = 50)	100% (0.95–1.00, n = 50)	
SD (L)	1.9% (0.00–0.09, n = 52)	26.9% (0.16–0.5, n = 52)	28.8% (0.18–0.42, n = 52)	12.0% (0.05–0.23, n = 50)	90.0% (0.80–0.96, n = 50)	15.4% (0.08–0.27, n = 52)	86.0% (0.75–0.94, n = 50)	100% (0.95–1.00, n = 50)	

For all the pathogens, multiple models fell within 2 ΔAICc units (hereafter “best models”) (Table 5), and all three explanatory variables were included within these models, except for FPV. However, based on the magnitude and direction of the parameter estimates, different variables showed unique associations with the prevalence of each pathogen (Table 6). For FeLV, the two best fit models (Akaike weight = 0.62 and 0.25) was composed of sex and CCA, and based on the estimates of both parameters, the prevalence of FeLV was significantly higher in high CCA districts and in females. Meanwhile, both of the two best models for the FPV prevalence with the highest AICc weights (0.39 and 0.33) included the CCA parameter. According to the parameter estimates, the prevalence of FPV in the low CCA districts is likely to be approximately three times higher than that of the high CCA districts; showing and opposite pattern with FeLV. For hemoplasma, sex was included in all three best models (Akaike weight = 0.35, 0.33, and 0.17), and the parameter estimates showing that males contracting hemoplasma infection would be approximately 2.23 times higher compare to females. Similarly, sex was the only variable included in all top three models (Akaike weight = 0.34, 0.26, and 0.13) for T. gondii; however, its relationship with prevalence was in the opposite direction for which females had three times higher odds of T. gondii infection. FCV was the only pathogen for which the PD parameter was consistently included in the three best models (Akaike weight 0.38, 0.31, and 0.16), and its prevalence showed a negative relationship with the PD. Regarding FHV, all top four models included the variable CCA; however, its influence on the FHV prevalence remains unverified based on its 95% confidence interval which included a zero and non-significant p-value (p = 0.99). In the model evaluation of the last two pathogens, FIV and B. henselae, none of the top models shared a variable. In the parameter estimates, 95% confidence intervals of all three variables included zero, and the variables did not show any significant influence on the infection status of FIV or B. henselae (Table 6).

Table 5 AICc values and AICc weights of the most parsimonious candidate models (models within 2 units of the ΔAICc) testing the effect of the explanatory variables (sex, population density (pd), and CCA) on the prevalence of the eight studied pathogens.

Pathogen	Model*	K	AICc	ΔAICc	AICc weight	
FeLV	sex + CCA	4	318.97	0.00	0.62	
sex + pd + CCA	5	320.80	1.84	0.25	
FPV	CCA	3	294.25	0.00	0.39	
sex + CCA	4	294.57	0.31	0.34	
Hemoplasma	sex + CCA	4	375.39	0.00	0.35	
sex	3	375.50	0.12	0.33	
sex + pd	4	376.86	1.48	0.17	
Toxoplasma gondii	sex + pd	4	159.77	0.00	0.34	
sex	3	160.29	0.52	0.26	
sex + CCA	4	161.64	1.87	0.13	
FCV	pd	3	117.76	0.00	0.38	
pd + sex	4	118.19	0.43	0.31	
pd + CCA	4	119.47	1.71	0.16	
FHV-1	pd + CCA	4	47.73	0.00	0.27	
CCA	3	47.81	0.09	0.25	
sex + pd + CCA	5	48.16	0.43	0.21	
sex + CCA	4	48.30	0.57	0.20	
FIV	Null	2	96.56	0.00	0.37	
pd	3	98.34	1.78	0.15	
CCA	3	98.48	1.93	0.14	
sex	3	98.52	1.96	0.14	
Bartonella henselae	Null	2	392.20	0.00	0.25	
CCA	3	392.85	0.65	0.18	
pd	3	392.98	0.78	0.17	
sex	3	393.87	1.67	0.11	
Note:

* pd, population density; CCA, cat caretaker activity.

Table 6 Model-averaged parameter estimates included in the most parsimonious models (models within 2 units of the ΔAICc) for each pathogen.

	Coefficient	S.E.	Upper CI	Lower CI	Odds ratio	p-Value	
FeLV							
CCA; low	−0.81	0.30	−1.42	−0.20	0.44	<0.01	
sex; male	−0.74	0.29	−1.32	−0.17	0.48	0.01	
pd	−0.00	0.01	−0.02	0.01	0.99	0.41	
FPV							
CCA; low	1.12	0.31	0.52	1.74	3.08	<0.01	
sex; male	−0.35	0.30	−0.93	0.23	0.70	0.24	
hemoplasma							
CCA; low	−0.37	0.25	−0.91	0.07	0.66	0.15	
sex; male	0.79	0.26	0.30	1.30	2.23	<0.01	
pd	−0.00	0.00	−0.01	0.01	0.99	0.67	
Toxoplasma gondii							
CCA; low	0.38	0.45	−0.50	1.27	1.47	0.40	
sex; male	−1.11	0.49	−2.07	−0.14	0.33	0.03	
pd	0.01	0.01	−0.00	0.02	1.01	0.08	
FCV							
CCA; low	0.38	0.82	−1.28	1.99	1.47	0.64	
sex; male	−0.63	0.55	−1.70	0.45	0.54	0.25	
pd	−0.02	0.01	−0.04	−0.01	0.98	<0.01	
FHV-1							
CCA; low	27.53	2701.87	−5268.04	5323.11	9.08 × 1011	0.99	
sex; male	−1.25	1.14	−3.49	0.99	0.29	0.28	
Pd	−0.18	0.13	−0.43	0.07	0.84	0.16	
FIV							
CCA; low	−0.14	0.65	−1.28	1.25	0.99	0.98	
sex; male	0.14	0.65	−1.25	1.28	1.01	0.98	
Pd	−0.00	0.01	−0.02	0.02	1.00	0.82	
Bartonella henselae							
CCA; low	−0.29	0.25	−0.77	0.19	0.75	0.24	
sex; male	−0.19	0.25	−0.67	0.29	0.83	0.44	
Pd	−0.00	0.00	−0.01	0.00	1.00	0.35	

Discussion

In contrast to our hypothesis, the PD of cats did not show a positive relationship with the intensity of the CCA. Instead, the PD of the cats was lower in the high CCA districts although the association was not statistically significant (Fig. 3). This trend may be explained by the difference in the proportion of sterilized cats among the districts as evidenced by the higher proportion of ear-marked cats in the high CCA districts in our observation. In certain districts, cat caretakers are known to heavily contribute to sterilizing stray cats in addition to the TNR done by the Seoul government as result of aggressive campaigning and education. The two high CCA districts (GN (H) and SC (H)) are the districts with the highest income in the city which is a socioeconomic factor that is known to be linked to either the proportion of sterilized stray cats or the level of sterilization by the caretakers (Finkler, Hatna & Terkel, 2011a, 2011b; Flockhart, Norris & Coe, 2016). This is also supported by the result of the cat caretaker survey done in the ROK (Kim et al., 2016), in which GN (H), SC (H), and MP (H) were among the top five districts regarding the proportion of cat caretakers with experience of sterilizing stray cats. Thus, although stray cats from the districts with a higher CCA may be exposed to more frequent and stable supplemental feeding, it is likely that more cats in these areas are also sterilized resulting in smaller population sizes compared to districts with a lower CCA.

From the eight studied pathogens, different associations were observed for each pathogen in relation to the PD and CCA. In three pathogens, FCV, FHV-1, and FeLV, with a transmission mode considered to rely on close contact of the host individuals (density-dependent transmission), we predicted its prevalence to be higher in a high CCA district due to higher PD. However, the pattern of association varied among these three pathogens. A significant link was observed between FeLV and CCA and between FCV and PD, whereas no single variable had a strong influence on explaining the prevalence of FHV-1. FCV and FHV-1 are two of the main pathogens causing feline respiratory tract disease and can especially be important in conditions where cats live socially. Compared to previous seroprevalence reports on pets or shelter cats (FCV, 36.6–92.4%; FHV-1, 11–73.3%) (Lappin et al., 2002; Lickey et al., 2005; DiGangi et al., 2012), results from our study were in a higher range for both FCV (95%) and FHV-1 (97.3%). However, Yamaguchi et al. (1996) reported high seroprevalence values for both FCV and FHV-1 (FCV, 100%; FHV-1, 100%) in a study of free-roaming farm cats, that shared supplemental food and latrine sites which is a condition perhaps more similar to our study setting compared to the other studies. Considering that both viral pathogens are highly contagious among cats in immediate proximity (Bannasch & Foley, 2005; Radford et al., 2009; Thiry et al., 2009), aggregation of individuals in close contact and/or higher survival rates of individuals after infection, driven by supplemental feeding (Lembo et al., 2011; Bitew et al., 2011; DiGangi et al., 2012), may have contributed to the high seroprevalence of FCV or FHV-1 observed in the study by Yamaguchi et al. (1996) and in our study. Nevertheless, the negative relationship of the FCV prevalence with PD was unexpected. We suspect that external variables may be involved in the observed relationship, such as a possibility of higher mortality in cats due to hypervirulent FCV infection in dense populations (Radford et al., 2007).

In the case of FeLV, its prevalence was significantly higher in districts with a high CCA (with GN and SC exhibiting the highest FeLV prevalence), but unrelated to the PD of stray cats. In comparison to previous studies on FeLV (2–23%) (Levy et al., 2006; Alves et al., 2011; Stojanovic & Foley, 2011), we observed a relatively high prevalence (12–42%), especially in two of our studied districts (GN (H) and SC (H)). Unlike FCV or FHV-1, transmission of FeLV relies less on aerosol or fomite, and its ability to survive outside the host is low (Lutz et al., 2009). Instead, it readily transmits through immediate amicable contacts such as food sharing, licking, and grooming (Rojko & Olsen, 1984; Barratt, 1997). We suggest that the higher FeLV prevalence in the two districts with a high CCA may be due to increased aggregation of cats around food provisioned by the cat caretakers. A recent simulation study demonstrated that increased aggregation and PD of supplemented cats led to a higher FeLV prevalence, although such impact varied depending on the influence of provisioned food on the immune response of cats (Becker & Hall, 2014). Considering the potential fatality of FeLV infection, we suggest further studies to clarify the long-term effect of providing supplemental food on stray cat populations.

For two pathogens with environmental transmission, T. gondii and FPV, we also predicted its prevalence to be positively associated with the PD of cats. In the case of T. gondii, the overall number of positive individuals was low in general (Table 3), similar to other studies on T. gondii prevalence in urban stray cats, which ranged from 5.61 to 14.3% (Lee et al., 2011; Wang et al., 2012; Oi et al., 2015). Additionally, its prevalence did not show any significant association with PD or CCA. Unlike feral cats in rural areas, where the prevalence of T. gondii in cats have been reported to be associated with the predation of intermediate host species (Afonso et al., 2007), transmission of T. gondii is more likely related to direct contact with oocysts in urban cat populations (Lelu et al., 2010). However, even within urban stray cats, fed populations are more likely to consume processed cat food, devoid of infectious T. gondii oocysts, while unfed populations would be more prone to rely on hunted prey that may function as intermediate hosts. Considering the potential risk that T. gondii may pose on public health as a zoonotic pathogen, we need to better understand the transmission dynamics of T. gondii between stray cats and those who are fed or interact closely with them on a daily basis.

The prevalence of FPV in our study ranged from 59% to 94%. The prevalence of FPV was highest in two districts with the lowest CCA (GC (L) and SD (L)) but showed a non-significant association with the host PD. FPV, the etiologic agent of feline panleukopenia, is a viral pathogen extremely resistant to external conditions enabling its environmental transmission by contact with various biotic or abiotic materials contaminated with fluids, fomites, or feces from an infected animal (Ikeda et al., 2002; Truyen et al., 2009). We expected the prevalence of FPV to be related to the accumulation of viral contaminants in the environment reflected by the PD in the area, but instead, the result indicated the association of FPV prevalence with CCA. Lack of food provisioning by caretakers in low CCA districts may have enabled stray cats in the area to maintain a larger activity range with a more intensive foraging activity compared to conspecifics in the high CCA districts with a relatively abundant and stable food source (Barratt, 1997; Tennent & Downs, 2008). This altered foraging strategy may lead to higher exposure rates of cats to pathogens transmitted by contact with environmental contaminants (Lepczyk, Lohr & Duffy, 2015). Similar associations have been discussed in other systems such as in the case of the increased infection to Strongyloides in wild primates (Parr, Fedigan & Kutz, 2013).

Among the three pathogens selected for their frequency-dependent transmission mode, FIV, B. henselae, and feline hemoplasma, none of the pathogens showed an association with CCA or PD as initially predicted. The low prevalence of FIV (total prevalence: 3.6%; 95% CI [0.018–0.049]) observed in this study has been commonly reported in other population-level studies in free-roaming or household cats (Lee et al., 2002; Levy et al., 2006; Ravi et al., 2010). FIV prevalence is most commonly explained in relation to the breeding ecology of free-roaming cats (Natoli et al., 2005; Fouchet et al., 2009). Urban stray cats are known to maintain a promiscuous mating system due to the high local density of both male and female cats (unlike its rural counterparts, wherein a polygynous mating system is observed (Pontier & Natoli, 1996)) which may lead to fewer aggressive interactions between males during breeding seasons and an overall low prevalence of FIV in urban male cats (Courchamp, Say & Pontier, 2000; Pontier et al., 2009; Ravi et al., 2010). In addition, neutering/spaying may have an additional impact on lowering the aggressive behavior related to breeding further influencing the prevalence of FIV in urban stray cats.

The prevalence of B. henselae also showed no association with the CCA or PD. However, considering that B. henselae is a zoonotic pathogen transmitted mostly through fleas, and that transmission of fleas can be facilitated by a high host PD (Krasnov, Khokhlova & Shenbrot, 2002), further attention is required regarding the prevalence of B. henselae in urban stray cats and its potential risk to cat caretakers. The prevalence of feline hemoplasma in male cats was higher than in female cats, similar to results from previous studies. It may be due to its transmission through aggressive sexual behavior (Walker Vergara et al., 2016; Bergmann et al., 2017). Overall, the spread of vector-borne and sexually transmitted diseases, such as B. henselae, feline hemoplasma, and FIV, are generally considered less impacted by the host PD or close contact among host individuals and is further supported by the results presented here.

To summarize, the findings from this study did not support our original hypothesis. First the PD and CCA did not show positive relationship, which was the fundamental assumption of our study. This shows that urban animal populations whose life-history is heavily interrupted by humans require consideration of multiple ecological and sociological factors to understand their population dynamics. Second, the pathogens considered to use “density-dependent transmission” modes, and originally expected to be positively related to CCA and host PD, such as FCV, FHV, FeLV, T. gondii, and FPV, mostly showed non-significant responses toward the two factors or mixed responses to CCA. Specifically, only FeLV and FPV showed significant relationships with CCA, but in different directions, with FeLV higher and FPV showing lower prevalence in higher CCA districts. This points out the possibility that in addition to the effect of supplemented food, which can increase immediate crowding of animals, its influence through other routes, such as altered foraging behavior, may also affect how these animals interact with their pathogens. However, study results may be limited by small sample size and the nature of convenience samples. Small sample size may have been especially problematic for pathogens with relatively high (FCV and FHV) or low prevalence (T. gondii and FIV) in throughout the studied districts. In addition, the samples used for pathogen testing were opportunistically collected through TNR centers, rather than targeted sampling. Due to the lack of exact overlap between cats surveyed for density analysis and cats used for pathogen testing, data used in this study may not be able to fully represent each other, although they were pooled and interpreted as samples from same district. Future similar studies may benefit from not only a larger sample size, but also by collecting samples specifically of the population where the density survey was performed.

With an increasing population size of stray cats in many urban cities, concerns arise regarding the impact of stray cats on the viability of wildlife populations and their potential role in disease transmission between humans as well as domestic and wild animals (Longcore, Rich & Sullivan, 2009; Lepczyk, Lohr & Duffy, 2015). In attempt to control the population size, sterilization is commonly adopted in many countries. However, it is often overlooked that sterilizing may also have an unexpected impact on the disease ecology of stray cat populations by shifting various behavioral traits of individuals and/or demographic structures (Finkler & Terkel, 2010; Finkler, Gunther & Terkel, 2011). For instance, sterilized cats may have a reduced aggressive or roaming behavior compared to intact cats and therefore have a reduced chance of exposure and transmission of infectious pathogens (Scott et al., 2002; Finkler & Terkel, 2010; Finkler, Gunther & Terkel, 2011). In contrast, pathogens transmitted through amicable behavior (e.g., conspecific grooming), such as FeLV, may be facilitated in populations where a high number of cats are sterilized (Jeon, 2011). Immigration of cats into neutered cat populations has been reported as a common occurrence (Gunther, Finkler & Terkel, 2011; Kilgour et al., 2017) with the potential risk of immigrants carrying and introducing pathogens. Currently, there is lack of concrete information to verify the epidemiological impact of sterilizing cats. Hence, additional efforts are warranted to clarify the impact of sterilization not only on the population control of stray cats but also on the health and welfare of free-roaming cats overall.

Conclusion

In conclusion, our findings did not meet our initial assumption that PD would be lower in high CCA districts. Likewise, the relationships between pathogen prevalence and PD or CCA varied for each tested pathogen rather than showing similar patterns depending on their transmission mode. Among the studied pathogens, only FPV and FeLV showed a significant association with the CCA but no association with PD. Our results suggest that supplemental feeding may influence disease ecology in the subject host population, not only by increasing the PD around the area as observed in previous studies on wildlife (Putman & Staines, 2004; Geisser & Reyer, 2005), but also by potentially changing the behavior or biology of animals, such as an altered aggregation and foraging behavior, which can affect their exposure and/or susceptibility to pathogens. So far, studies on urban stray cats with an emphasis on health aspects have been mostly limited to screening for the prevalence of zoonotic pathogens. However, we suggest that the incorporation of multifaceted information, such as human behaviors and perceptions about urban stray cats, the influence of human behaviors on the ecology of cats and the transmission of pathogens of interest, would be crucial to better understand the potential epidemiological risk that urban stray cats may pose to wildlife and/or human public health and to design efficient management strategies.

Supplemental Information

Supplemental Information 1 Fig. S1. Typical landscape of residential area of Seoul where the density survey was performed.

Photo by Jusun Hwang.

Click here for additional data file.

Supplemental Information 2 Raw dataset.

Click here for additional data file.

We sincerely thank Daniel Becker, Dr. Jefferey Hepinstall-Cymerman, Dr. Annie Page-Karjian and the anonymous reviewers for their insightful comments on the earlier versions of the manuscript.

Additional Information and Declarations

Competing Interests

Author Contributions

Data Availability

1 We modified a source figure produced by “Stefan-Xp” under the following copyright permission. “Permission is granted to copy, distribute and/or modify this document under the terms of the GNU Free Documentation License, Version 1.2 or any later version published by the Free Software Foundation; with no Invariant Sections, no Front-Cover Texts, and no Back-Cover Texts. A copy of the license is included in the section entitled the GNU Free Documentation License. The GNU Free Documentation License (GNU FDL or simply GFDL) is a copyleft license for free documentation, designed by the Free Software Foundation (FSF) for the GNU Project.”

The authors declare that they have no competing interests.

Jusun Hwang conceived and designed the experiments, performed the experiments, analyzed the data, prepared figures and/or tables, authored or reviewed drafts of the paper.

Nicole L. Gottdenker conceived and designed the experiments, analyzed the data, prepared figures and/or tables, authored or reviewed drafts of the paper, approved the final draft.

Dae-Hyun Oh conceived and designed the experiments, performed the experiments, analyzed the data.

Ho-Woo Nam performed the experiments, analyzed the data.

Hang Lee contributed reagents/materials/analysis tools, authored or reviewed drafts of the paper, approved the final draft, provided laboratory to perform the study.

Myung-Sun Chun contributed reagents/materials/analysis tools, authored or reviewed drafts of the paper, approved the final draft, provided funding for the study.

The following information was supplied regarding data availability:

The raw data are provided in a Supplemental File.

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
