# Peer review of "Disentangling the link between supplemental feeding, population density, and the prevalence of pathogens in urban stray cats"

_PeerJ, doi:10.7717/peerj.4988_

## Round 0.1 · original submission · Minor Revisions

Your article has been carefully reviewed, and both reviewers identify many strengths and interesting novel findings, but also and some areas requiring clarification or additional information. In particular, please note the concern regarding incomplete availability of raw data and needed additional details regarding the analysis.

Reviewer 1 ·

Basic reporting

I have enjoyed reviewing this paper as submitted to PeerJ. It is well written and has a very interesting hypothesis that was tested using a combination of field data collection, and biological sampling of free-roaming cats. My comments and suggestions are focused on improving the clarity of the paper and on adding additional context for some specific sections of the text. The paper in its current format meets the PeerJ requirements regarding Basic reporting.

In general the flow and clarity could be improved by paying particular attention to sentence length. Many sentences are very long and although they are clear in terms of language the writing could be improved by breaking these into shorter sentences. E.g. line 88-92 1 sentence, 5 lines, line 77-82, 1 sentence, 6 lines

The authors switch between using the following terms, density, crowding, and concentration in the Introduction. These terms appear to be used as substitutes for one another however, I would recommend that it would be more accurate to identify a single, well defined definition and then use that in the text. For instance, different individuals or host-pathogen systems may define crowding in different ways and in the use here, it is less clear what exactly is intended.

The paragraph that begins at line 86 – this section which speaks to the differences between density-dependent and frequency dependent transmission would really benefit from a paragraph that could come before this to introduce more generally the fundamental and theoretical framework regarding these 2 “types” of pathogen transmission dynamics. The paragraph starting at line 86 provides some excellent examples but a more general lead in would add more context and improve the clarity.

Line 96 – here you seem to be saying that contact rates among host individuals are not an important component of frequency dependent transmission. I would argue the opposite, that in fact it is contact rates (sometimes artificially inflated by behaviour) that create the enhanced opportunities for disease transmission in this case.

Line 126-129 – this sentence implies that these species are perhaps endangered. Is there a specific designation that should be mentioned regarding this species? I would also expand upon the specifics regarding why free-roaming cat contact could have important consequences for these populations? Death due to pathogens? If this is the case, then it might be wise to pull in some of the literature here regarding the way in which different “modes” of pathogen transmission might influence species persistence vs. extinction. If food provisioning causes aggregation that increases the risk of transmission via frequency-dependent contacts than pathogen -mediated extinction would be a real possibility (but it is not entirely clear if this is where you were going with this sentence).

Line 140 – Just a note that density-dependent transmission does not exclusively mean direct contact transmission. Sexual transmission is also direct contact transmission. The difference is that behaviorally, even if hosts are rare, individuals will seek out sexual contacts over wider ranges which means that transmission occurs independent of density. In this case, transmission is driven by “frequency of infected contacts”, which means prevalence of the infection. If prevalence amongst contacts is high then you have more transmission. I have always thought that it would be clearer to use density-dependent and density- independent transmission as the terminology but so far no one seems to have taken up this idea 

Line 148 – the environmental reservoir bit here overrides the direct transmission hypotheses. I would perhaps elaborate on why this is assumed to be positively associated with population density (e.g. is it really shedding into the environment creating a point source exposure?)

I think it would be very helpful to have a summary table of the different pathogens that were the focus of the seroprevelance data with a column for the “mode of transmission” and then some other summary measures. I find it very difficult to read through the text with all of the FCV/FHV/FeLV in the results section and try to keep straight which pathogen is which, which one is transmitted in which way, and how this is tied to your hypothesis about density dependent and density independent transmission. Not a requirement but I do think it might be helpful.

Experimental design

The experimental design meets the requirements of the PeerJ review policy.

Line 176 – the sample size is missing here. N=0000
Line 187 – 190 – this seems to be a result (Table 1) of the PCA rather than a method.
Line 197 – Need to define SP here. This also comes a bit out of the blue. If this was one of the top four than it should be listed alongside the other three and then the justification re: exclusion is clearer, and the reader then sees that you are only using 3 of the top 4 high CCA sites.
Figure 2 – I am not sure of the PeerJ protocol for referencing this sort of figure so will leave it up to the editor to identify if this is appropriate.
Line 217 – change this sentence to say, “For logistical reasons…
Line 224- change this sentence to say “..used for the density survey consisted…(remove the word is)
Lines 250-251 – this is just a thought but I find it difficult to remember which of the site abbreviations is associated with each of the CCA levels. Might it be reasonable for clarity to label the sites as high-site 1, high-site 2, low-site 1, low-site 2 etc. or something similar throughout the text which would make it easier to keep track as the reader moves through the paper? You have done this in the Tables/Figures, but the text would benefit from a similar approach.
Line 266 – I think this must be a typo but this sentence seems to say that all of the cats were determined to be too young for sterilization and therefore were released. I don’t think that is actually what you mean.
Line 294 – The Chong et al. reference here is duplicated
Line 309 - Just to be clear, capture-recapture was done just based on a visual survey that was handwritten on a data form? I am curious why you did not use other techniques for resighting cats such as by using photos etc. I think that something that is missing in your Conclusion is some discussion of your study limitations in regard to the methodology.
Line 347 – I am a bit unclear how CCA is an explanatory variable for sterilization rates. If this is the case, then it should be more clearly discussed in the intro if there is some known association between these two things.

Validity of the findings

The model findings are valid and interesting. I would suggest the inclusion of a section that speaks to the study limitations as that is not currently specifically included.

Line 391-412 - While I recognize that the results are in the Tables, it would be helpful in the results section where you are talking about the “best” models to perhaps indicate something about the “best” interpretation but putting the AIC in brackets or some measure to help the reader follow along.

Lines 422 –437 – I think this is a reasonable interpretation of the findings which are counterintuitive. I also wonder if it might be the case, that high CCA sites just have lots more people feeding so the cats are more dispersed among the feeding sites making their density less on the whole? Whereas in low CCA sites, because there are few people feeding the cats they congregate more into smaller areas.

Line 551 – I think here that it is more appropriate to state that your findings did not support your original hypothesis.

Additional comments

Line 86 – replace with, “…interactions as a result of host population density..
Line 78 – replace with, “…crowding of host individuals has been…”
Line 89 – replace with, “…likely to be facilitated as host population density increases, as with Brucella…”
Line 90 – replace with, “..success of the pathogens transmission…”
Line 113 –replace with, “…centered around free-roaming cats…”
Line 180 – replace with, “ …in a regular manner..”
Line 336 – replace with, “…50 meters was…”
Line 344 – replace with, “…report..”
Line 365 – remove, “and respectively”
Line 371 – replace with, “…compared…”
Line 400 – replace with, “…showing …”
Nowhere is there any discussion of the sample size related to the blood samples collected. I am wondering if it is possible that some of the results might be a function of just having a small number of convenience blood samples rather than a more targeted sampling protocol that was linked to supplemental feeding sites specifically. I am aware of how difficult that would be but it is worth at least some discussion of the sample size in light of the findings.
Figure 1 – I would tend to label the points in black text rather than red. In the figure caption I would also speak to the point that one of the high CCA sites was excluded.
Figure 3 – I am not sure about the rationale for the order of the x-axis on the figure, but I think it would be easier to compare if all of the sites e.g. DDM1 and DDM2 were side by side. This is true for only some of the sites on the figure.
Table 2 – Need a space between CI and cats in the column header.
Table 4- Again, I would find it helpful to have the abbreviations spelled out here either as a footer, in the body of the table or in the Table heading.

Reviewer 2 ·

Basic reporting

The raw data file does not appear to be complete. There is no description of what the variable titles mean (eg. pd), nor is there any code provided for analyses. Data on variables included in this study, including any data concerning pathogen abundance, have not been provided.

It seems to me that the hypothesis of this study is that supplemental feeding leads to increased population density due to the increase of a resource (food) that would otherwise be limiting their populations, and that this has consequences for disease transmission. From this, the authors predict that increased food availability (in high CCA areas) will show increased population density, and also increased transmission of density-dependent pathogens (FCV, FeLV, and FHV-1). They also predict that frequency-dependently transmitted pathogens (FIV, feline hemoplasma, and B. henselae) will not be affected by differences in population density. The distinction between hypotheses and predictions are sometimes well distinguished (lines 422-423), but needs to be clarified in other areas (lines 33-36, 149).

If any special R packages were used in the analyses, please list and cite them.

There is an extensive description of why site SP was not included in the analysis as a high CCA site. However, there is no parallel description of why site GB was not included as a low CCA site. Please provide some information, since this site is highlighted in Figure 1.

This manuscript provides an impressive amount of data to the reader, and it is generally well summarized in tables and figures. There are a few places where additional data summaries would be helpful and I have included them in the line number listing below.

Line number and suggested corrections:
27: “feral or stray animals”
40-41: “Blood samples (n=302) were collected from stray cats by local animal hospitals…”
65: “..urban trash cans and pets fed outdoors)…”
69: “…to pathogen host community…”
113: “..around free-roaming cats, as the availability..”
131: “…pathogen prevalence are associated in urban stray cats…”
136: “…Toxoplasma gondii. Each of these pathogens has a different transmission mode, falling…”
174: “…the number of cat caretakers and the number of cats cared for per caretaker.”
176: What does (n=0000) refer to?
176: “…a 2016 nationwide cat-caretaker questionnaire (Kim et al. 2016)…”
183: I don’t understand the last part of this sentence (“in the total number of respondents from each district”). Is this another (the 7th?) question on the questionnaire?
217: “Due to logistical reasons…”
218: “…was conducted in the fall…”
220: When describing the transect lengths, please provide the mean and the range of the transects in the text.
223: From description in other parts of the manuscript, it is clear that each transect was walked 5 times. Please clarify the language in the Field Survey section to indicate that each transect was surveyed 5 times each (e.g. “ Each transect was surveyed 5 times, once per day for 5 days, within a 2 week period.”)
226: “…to thoroughly cover the blocks, as in methods used in previous studies…”
235: “…two meters of range, close enough to identify…”
336: The metric conversion of 0.3ha is 3000m2, which could be 55m x 55m or it could be 1m x 3000m. The justification for the buffer zone could be clarified a bit.
344: “We calculated the mean and standard deviation…”
361: Please list the average number of cats observed across sites in addition to the range.
382: “..of the pathogens: T. gondii, …”
409: “…and non-significant p-value…”
446: “…conditions where cats live socially.”
465: “…we observed a relatively high prevalence (12-42%),…”
500: “…districts may have enabled stray cats in the area…”
551: “..meet our initial assumption that population density would be lower…”

Experimental design

No comment.

Validity of the findings

The raw data file does not appear to be complete. There is no description of what the variable titles mean (eg. pd), nor is there any code provided for analyses. Data on variables included in this study, including any data concerning pathogen abundance, have not been provided.

Additional comments

This timely and important paper offers new insight into the current conversation on urban, free-roaming cats. This study assesses the prevalence of 8 pathogens in 6 districts across two urban environment types: high and low Cat Caretaker Activity. Taking place in a highly urban center, the authors have done an excellent job at describing methods and reporting the findings of pathogen prevalence, and have very well coordinated a study that combines aspects of wildlife ecology and pathobiology. The interpretation of the results is well-founded and provides an excellent jumping-off point for future studies on the impacts of (zoonotic) feline pathogen risk in urban environments. The hypotheses and predictions, although repeatedly described throughout (which is great), are a bit mixed up in terms of how they are identified. Additionally, there are some linguistic concerns, listed below, and some incomplete information provided. Pending these minor changes, I recommend this paper for publication.

---

## Round 0.2 · accepted · Accept

You have addressed the reviewers' previous comments in detail and thoroughly, and the manuscript now offers a well written contribution to the literature concerning population dynamics in stray cats.

#